# Learning Representation for Bayesian Optimization with Collision-free Regularization

## Abstract

Bayesian Optimization has been challenged by the large-scale and high-dimensional datasets, which are common in real-world scenarios. Recent works attempt to handle such input by applying neural networks ahead of the classical Gaussian process to learn a (low-dimensional) latent representation. We show that even with proper network design, such learned representation often leads to collision in the latent space: two points with significantly different observations collide in the learned latent space, leading to degraded optimization performance. To address this issue, we propose LOCo, an efficient deep Bayesian optimization framework which employs a novel regularizer to reduce the collision in the learned latent space and encourage the mapping from the latent space to the objective value to be Lipschitz continuous. LOCo takes in pairs of data points and penalizes those too close in the latent space compared to their target space distance. We provide a rigorous theoretical justification for LOCo by inspecting the regret of this dynamic-embedding-based Bayesian optimization algorithm, where the neural network is iteratively retrained with the regularizer. Our empirical results further demonstrate the effectiveness of LOCo on several synthetic and real-world benchmark Bayesian optimization tasks.

## 1 Introduction

Bayesian optimization is a classical sequential optimization method and is widely used in various fields in science and engineering, including recommender systems (Galuzzi et al., 2019), medical trials (Sui et al., 2018), robotic controller optimization (Berkenkamp et al., 2016), scientific experimental design (Yang et al., 2019), and hyper-parameter tuning (Snoek et al., 2012), among many others. Many of these applications involve evaluating an expensive blackbox function; therefore, the number of queries should be minimized. A common way to model the unknown function is via Gaussian processes (GPs) (Rasmussen and Williams, 2006), which have been extensively studied under the bandit setting, as an effective surrogate model of the unknown objective function in a broad class of blackbox function optimization problems (Srinivas et al., 2010; Djolonga et al., 2013).

A key computational challenge for learning with GPs lies in optimizing specific kernels used for modeling the covariance structures. Such an optimization task depends on the dimension of the input space. For high-dimensional data, it is often prohibitive to train a GP model. Meanwhile, local kernel machines are known to suffer from the curse of dimensionality (Bengio et al., 2005), while the required number of training samples could grow exponentially with the dimensionality of the data. Therefore, dimensionality reduction and representation learning algorithms are needed to optimize the learning process.

Recently, Gaussian process optimization has been investigated in the context of latent space models. For example, deep kernel learning (Wilson et al., 2016) learns a (low-dimensional) data representation and a scalable kernel simultaneously via an end-to-end trainable deep neural network. In general, the neural network is trained to learn a simpler latent representation with reduced dimension and has the structure information already embedded for the GP. Combining the representation learned via a neural network with GP could improve the scalability and extensibility of classical Bayesian optimization, but it also poses new challenges for the optimization task, such as dealing with the tradeoff between representation learning and function optimization (Tripp et al., 2020).

As we later demonstrate, a critical challenge brought by introducing representation learning into Bayesian optimization is that the latent representation is prone to *collisions*: two points with significantly different observations can get too close, and therefore collide in the latent space. The collision effect in latent space models for Bayesian optimization is especially evident when information is lost during dimensionality reduction and/or when the training data is limited in size. As illustrated in Figure 1, when passed through the neural network, data points with drastically different observations are mapped to close positions in the latent space. Such collisions could be regarded as additional noise introduced by the neural network. Although Bayesian optimization is known to be robust to mild noisy observations (Bogunovic et al., 2018), the collision in latent space could be harmful to the optimization performance, as it is non-trivial to model the collision into the acquisition function explicitly. Also, the additional noise induced by the collision effect will further loosen the regret bound for classical Bayesian optimization algorithms (Srinivas et al., 2010).

**Overview of main results**    To mitigate the collision effect, we propose a novel regularization scheme that can be applied as a simple plugin amendment for the latent space based Bayesian optimization models. The proposed algorithm, namely *Latent Space Optimization via Collision-free regularization* (LOCo), leverages a regularized regression loss function to optimize the latent space for Bayesian optimization periodically.

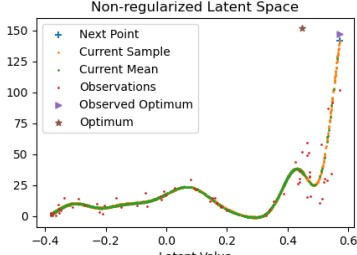

Concretely, our collision-free regularizer is encoded by a novel *pairwise collision penalty* function defined jointly on the latent space and the output domain. In order to mitigate the risk of collision in the latent space (and consequently boost the optimization performance), LOCo applies the regularizer to minimize the collisions uniformly in the latent space.

We further note that for Bayesian optimization tasks, collisions in regions close to the optimum are more likely to mislead the optimization algorithm. Based on this insight, we propose an optimization-aware regularization scheme that assigns higher weight to the collision penalty on those pairs of points closer to the optimum region in the latent space. This algorithm, which we refer to as *Dynamically-Weighted LOCo*, is designed to dynamically assess the importance of a collision during optimiza-

Figure 1: The collision effect in latent space-based BO tasks when having 100 data points and observations. Because the data points around the optimum severely collided, BO is misguided to the sub-optimum. Different from noise which normally assumed to be Gaussian, the collision could be much more divergent. See extended discussion in Appendix B.1.

tion. Compared with the uniform collision penalty in the latent space, the dynamic weighting mechanism has demonstrated drastic improvement over the state-of-the-art latent space based Bayesian optimization models.

We summarize our key contributions as follows:

I. We investigate latent space based Bayesian optimization, and expose the limitations of existing latent space optimization approaches due to the collision effect on the latent space (Section 3).

II. We propose a novel regularization scheme as a simple plugin amendment for latent space based Bayesian optimization models. Our regularizer penalizes collisions in the latent space and effectively reduces the collision effect. Furthermore, we propose an optimization-aware dynamic weighting mechanism for adjusting the collision penalty to improve the effectiveness of regularization for Bayesian optimization (Section 4).

III. We provide theoretical analysis for the performance of Bayesian optimization on regularized latent space (Section 5).

IV. We conducted an extensive empirical study on several synthetic and real-world datasets, including a real-world case study for cosmic experimental design, and demonstrate the promising empirical performance for our algorithm (Section 6).

## 2 RELATED WORK

This section provides a short survey on recent work in Bayesian learning, which was designed to overcome the high-dimensionality challenge for Gaussian process regression tasks and Bayesian optimization.

**Different surrogate models with internal latent space**   Some alternative surrogate models have been proposed to replace classical GP in Bayesian optimization to overcome the challenge of high-dimensional and highly-structured input in BO. *Deep Network for Global Optimization* (DNGO) Snoek et al. (2015) uses a pre-trained deep neural network with a Bayesian linear regressor at the last hidden layer of the network as the surrogate model. More generally, *Deep Kernel Learning* (DKL) combines the power of the Gaussian process and neural network by introducing a deep neural network $g$ to learn a mapping $g : \mathcal{X} \to \mathcal{Z}$ from the input domain $\mathcal{X}$ to a latent space $\mathcal{Z}$ (Wilson et al., 2016). It uses the latent representation $z \in \mathcal{Z}$ as the input of the base Gaussian process. The neural network $g$ and a spectral mixture-based kernel $k$ form a scalable expressive closed-form deep covariance kernel, denoted by $k_{DK}(x_i, x_j) \to k(g(x_i), g(x_j))$. The deep kernel allows end-to-end learning and Bayesian optimization on the original input space.

**Representation learning and latent space optimization**   Instead of reducing the dimensionality and performing optimization in an end-to-end process, other methods aim to optimize in a related low-dimensional space first and then map the solution back to the original input space. Djolonga et al. (2013) assume that only a subset of input dimensions varies, and the kernel is smooth (i.e. with bounded RKHS norm). Under these assumptions, the underlying subspace is learned via low-rank matrix recovery. *Random feature* is another solution under this setting (Rahimi et al., 2007; Letham et al., 2020; Binois et al., 2015; Nayebi et al., 2019; Wang et al., 2016). It is known that a random representation space of sufficiently large dimension is guaranteed to contain the optima with high probability. Mutný and Krause (2019) consider *Quadrature Fourier Features* (QFF)—as opposed to *Random Fourier Feature* (RFF) in Rahimi et al. (2007)—to overcome the variance starvation problem, and proved that Thompson sampling and GP-UCB achieve no-regret with squared exponential kernel in optimization tasks. However, both RFF and QFF methods rely on a key assumption that the function to be optimized has a low *effective dimension*. In contrast, as discussed in Section 6 and the supplemental materials, we show that LOCo performs well for challenging high-dimensional BO problems where algorithms relying on the low effective dimension assumption may fail.

Another line of work on latent space optimization uses *autoencoders* to learn low-dimensional representations of the inputs to improve the scalability and capability to leverage the structural information (Mathieu et al., 2019), (Ding et al., 2020), (Gómez-Bombarelli et al., 2018; Huang et al., 2015; Tripp et al., 2020; Lu et al., 2018). Mathieu et al. (2019), Ding et al. (2020) focus on disentangled representation learning that breaks down, or disentangles, each feature into narrowly defined variables and encodes them as separate dimensions.Tripp et al. (2020) iteratively train the autoencoder with a dynamic weighting scheme when performing optimization to improve the embedding. Griffiths and Hernández-Lobato (2020) and Letham et al. (2020) enforce certain properties on the representation space to improve the optimization performance. To the best of the authors' knowledge, collision of the embeddings has not been explicitly studied. Binois et al. (2015) propose a *warped kernel* to guarantee the injectivity in the random linear embedding, which is not applicable in neural network-based methods.

A common challenge in applying these techniques to generic optimization tasks lies in the assumption on the accessibility of training data: Bayesian optimization often assumes limited access to labeled data, while surrogate models built on deep neural networks often rely on abundant access to data for pretraining. Another problem lies in the training objective: During the training phase, these surrogate models typically focus on improving the *regression* performance, and do not explicitly address the artifact caused by collisions of the learned embeddings, which—as we later demonstrate in Section 3.3—could be harmful to sequential decision-making tasks.

## 3 PROBLEM STATEMENT

In this section, we introduce necessary notations and formally state the problem. We focus on the problem of sequentially optimizing a function $f : \mathcal{X} \to \mathbb{R}$, where $\mathcal{X} \subseteq \mathbb{R}^d$ is the input domain. At iteration $t$, we pick a point $x_t \in \mathcal{X}$, and observe the function value perturbed by additive noise: $y_t = f(x_t) + \epsilon_t$ with $\epsilon_t \sim \mathcal{N}(0, \sigma^2)$ being i.i.d. Gaussian noise. Our goal is to maximize the sum of rewards $\sum_{t=1}^{T} f(x_t)$ over $T$ iterations, or equivalently, to minimize the *cumulative regret* $R_T := \sum_{t=1}^{T} r_t$, where $r_t := \max_{x \in \mathcal{X}} f(x) - f(x_t)$ denotes the *instantaneous regret* of $x_t$.

## 3.1 BAYESIAN OPTIMIZATION

Formally, we assume that the underlying function $f$ is drawn from a Gaussian process $\mathcal{GP}(m(x), k(x, x'))$, where $m(x)$ is the mean function and $k(x, x')$ is the covariance function. At iteration $t$, given the selected points $A_t = \{x_1, \ldots, x_t\}$ and the corresponding noisy evaluations $\mathbf{y}_t = [y_1, \ldots, y_t]^\top$, the posterior over $f$ also takes the form of a GP, with mean $\mu_t(x) = k_t(x)^\top (K_t + \sigma^2 I)^{-1} \mathbf{y}_t$, covariance $k_t(x, x') = k(x, x') - k_t(x)^\top (K_t + \sigma^2 I)^{-1} k_t(x')$, and variance $\sigma_t^2(x) = k_t(x, x)$, where $k_t(x) = [k(x_1, x), \ldots, k(x_t, x)]^\top$ and $K_t$ is the positive definite kernel matrix $[k(x, x')]_{x, x' \in A_t}$ (Rasmussen and Williams, 2005). After obtaining the posterior, one can compute the acquisition function $\alpha : \mathcal{X} \to \mathbb{R}$, which is used to select the next point to be evaluated. Various acquisition functions have been proposed in the literature, including popular choices such as Upper Confidence Bound (UCB) (Srinivas et al., 2010) and Thompson sampling (TS) (Thompson, 1933).

**Remark.** *Regret is commonly used as performance metric for BO methods. In this work we focus on the simple regret $r_T^* = \max_{x \in \mathcal{X}} f(x) - \max_{t \leq T} f(x_t)$ and cumulative regret $R(T) = \sum_{t=1}^T r_t$.*

## 3.2 LATENT SPACE OPTIMIZATION

Recently, *Latent Space Optimization* (LSO) has been proposed to solve Bayesian optimization problems on complex input domains (Gómez-Bombarelli et al., 2018; Huang et al., 2015; Tripp et al., 2020; Lu et al., 2018). LSO learns a latent space mapping $g : \mathcal{X} \to \mathcal{Z}$ to convert the input space $\mathcal{X}$ to the latent space $\mathcal{Z}$. Then, it constructs an objective mapping $h : \mathcal{Z} \to \mathbb{R}$ such that $f(x) \approx h(g(x))$, $\forall z \in \mathcal{Z}$. In this paper, we model the latent space mapping $g$ as a neural network; the neural network $g$ and the base kernel $k$ together are regarded as a *deep kernel*, denote by $k_{\mathrm{nn}}(x, x') = k(g(x), g(x'))$ (Wilson et al., 2016). In this context, the actual input space for BO is the latent space $\mathcal{Z}$ and the objective function is $h$. With the acquisition function $\alpha_{\mathrm{nn}}(x) := \alpha(g(x))$, we do not compute an inverse mapping $g^{-1}$ as opposed to the aforementioned autoencoder-based LSO algorithms (e.g. Tripp et al. (2020)), since BO directly select $x_t = \arg\max_{x \in \mathcal{X}} \alpha_{\mathrm{nn}}(x) \; \forall t \leq T$. In our analysis, we use squared exponential kernel, i.e. $k_{\mathrm{SE}}(x, x') = \sigma_{\mathrm{SE}}^2 \exp\left(-\frac{(x-x')^2}{2l}\right)$.

## 3.3 THE COLLISION EFFECT OF LSO

When the mapping $g : \mathcal{X} \to \mathcal{Z}$ is represented by a neural network, it may cause undesirable *collisions* between different input points in the latent space $\mathcal{Z}$. Under the noise-free setting, we say there exists a *collision* in $\mathcal{Z}$, if $\exists x_i, x_j \in \mathcal{X}$, such that when $g(x_i) = g(x_j)$, $|f(x_i) - f(x_j)| > 0$. Such collision could be regarded as additional (unknown) noise on the observations introduced by the neural network $g$. Given a representation function $g$, noisy observations $y = f(x) + \epsilon$, we say that there exists a collision, if for $\lambda > 0$, there exist $x_i, x_j \in \mathcal{X}$, such that $|g(x_i) - g(x_j)| < \lambda |y_i - y_j|$.

When the distance between a pair of points $(x_i, x_j)$ in the latent space is too close compared to their difference in the output space, the different output values $y_i, y_j$ for the collided points in the latent space could be interpreted as the effect of additional observation noise for $g(x_i)$ (or $g(x_j)$). In general, collisions could degrade the performance of LSO. Since the collision effect is *a priori* unknown, it is often challenging to be dealt with in LSO, even if we regard it as additional observation noise and increase the (default) noise variance in the Gaussian process. Thus, it is necessary to mitigate the collision effect by directly restraining it in the representation learning phase. One potential method to avoid collision could be tuning the design of neural networks. However, we empirically show that increasing the network complexity often does *not* help to reduce the collision. The study is posed in Appendix B.2.

We consider a low-noise setting where the collision can play a more significant role in degrading the optimization performance. As is shown in Figure 1 that the collision could result in larger difficulty in the optimization task. And when a collision exists, it is hard to distinguish it from the observation. Therefore we focused on treating the collision when defining the penalty instead of dealing with the stochasticity.

# 4 LATENT SPACE OPTIMIZATION VIA COLLISION-FREE REGULARIZATION

We now introduce LOCo as a novel algorithmic framework to mitigate the collision effect.

## 4.1 OVERVIEW OF THE LOCo ALGORITHM

The major challenge in restraining collisions in the latent space is that—unlike the formulation of the classical regression loss—we cannot quantify it based on a single training example. We can, however, quantify collisions by grouping pairs of data points and inspecting their corresponding observations.

We define the *collision penalty* based on pairs of inputs and further introduce a pair loss function to characterize the collision effect. Based on this pair loss, we propose a novel regularized latent space optimization algorithm[1], as summarized in Algorithm 1. The proposed algorithm concurrently feeds the pair-wise input into the same network and calculates the pair loss function. We demonstrate this process in Figure 2.

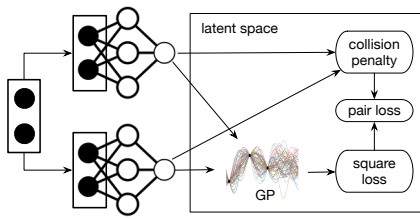

Given a set of labeled data points, we can train the neural network to create an initial latent space representation similar to DKL (Wilson et al., 2016)[2]. Once provided with the initial representation, we can then refine the la-

Figure 2: Schematic of LOCo

tent space by running LOCo and periodically update the latent space (i.e. updating the learned representation after collecting a batch of data points) to mitigate the collision effect as we gather more labels.

---

**Algorithm 1** **L**atent Space **O**ptimization via **Co**llision-free Regularization (LOCo)

1: **Input**: Penalty parameter $\lambda$ (cf. Equation 1), regularization weight $\rho$ (cf. Equation 3), importance weight parameter $\zeta$ (cf. Equation 2), neural network $g$, parameters $\theta_t = (\theta_{h,t}, \theta_{g,t})$, total time steps $T$;
2: **for** $t = 1$ *to* $T$ **do**
3:   $x_t \leftarrow \arg\max_{x \in D} \alpha(g(x, \theta_{g,t}))$        ▷ *maximize acquisition function*
4:   $y_t \leftarrow$ evaluation on $x_t$             ▷ *update observation*
5:   $\theta_{t+1} \leftarrow$ retrain $\theta_t$ with the pair loss function $L_{\rho,\lambda,\zeta,g}(\theta_t, D_t)$ as in Equation 3
6: **Output**: $\max_t y_t$

---

## 4.2 COLLISION PENALTY

This subsection aims to quantify the collision effect based on the definition proposed in Section 3.3. As illustrated in Figure 2, we feed pairs of data points into the neural network and obtain their latent space representations. Apart from maximizing the GP's likelihood, we concurrently calculate the amount of collision on each pair and incur a penalty when the value is positive. For $x_i, x_j \in \mathcal{X}$, $y_i = f(x_i) + \epsilon, y_j = f(x_j) + \epsilon$ are the corresponding observations, and $z_i = g(x_i), z_j = g(x_j)$ are the corresponding latent space representations. We define the *collision penalty* as

$$p_{ij} = \max(\lambda|y_i - y_j| - |z_i - z_j|, 0) \tag{1}$$

where $\lambda$ is a penalty parameter that controls the smoothness of the target function $h : \mathcal{Z} \to \mathbb{R}$. As a rule of thumb, one can estimate $\lambda$ by sampling from the original data distribution $\mathbb{P}(X, Y), (X, Y) \in \mathcal{X} \times \mathbb{R}$, and choose the $\lambda$ to be the maximum value such that $\sum_{i,j} \max(\lambda|y_i - y_j| - |x_i - x_j|, 0) = 0$ (i.e. to provide an upper bound for $\lambda$ by keeping the total collision in the input domain to be zero).

---

[1]Note that we have introduced several hyper-parameters in the algorithm design; we will defer our discussion on the choice of these parameters to Section 6.

[2]To obtain an initial embedding in the latent space, the process does not require the labels to be exact and allows the labels to be collected from a related task of cheaper cost.

### 4.3 IMPORTANCE-WEIGHTED COLLISION-FREE REGULARIZER

Note that it is challenging to universally reduce the collisions by minimizing the collision penalty and the GP's regression loss—this is particularly the case with a limited amount of training data. Fortunately, for optimization tasks, it is often unnecessary to learn fine-grained representation for suboptimal regions. Therefore, we can dedicate more training resources to improve the learned latent space pertaining to the potentially near-optimal regions. Following this insight, we propose to use a weighted collision penalty function, which uses the objective values for each pair as an importance weight in each iteration. Formally, for any pair $((x_j, z_j, y_j), (x_i, z_i, y_i))$ in a batch of observation pairs $D_t = \{((x_m, z_m, y_m), (x_n, z_n, y_n))\}_{m,n}$ where $x_n, x_m \in A_t$ and $y_n, y_m \in \mathbf{y}_t$, we define the *importance-weighted penalty function* as

$$\tilde{p}_{ij} = p_{ij} w_{ij} \qquad \text{with} \qquad w_{ij} = \frac{e^{\zeta(y_i + y_j)}}{\sum\limits_{(m,n) \in D_t} e^{\zeta(y_m + y_n)}}. \tag{2}$$

Here the importance weight $\zeta$ is used to control the aggressiveness of the weighting strategy.

Combining the kernel learning objective—negative log likelihood and the collision penalty for GP, we define the *pair loss* function $L_{\rho,\lambda,\zeta,g}$ as

$$L_{\rho,\lambda,\zeta,g}(\theta_t, D_t) = -\log(P(\mathbf{y}_t | A_t, \theta_t)) + \frac{\rho}{||D_t||^2} \sum_{i \in D_t, j \in D_t} \tilde{p}_{ij} \tag{3}$$

Here, $-\log(P(\mathbf{y}_t | A_t, \theta_t)) = -\frac{1}{2}\mathbf{y}_t^\top (K_t + \sigma^2 I)^{-1} \mathbf{y}_t - \frac{1}{2}|(K_t + \sigma^2 I)| - \frac{t}{2}\log(2\pi)$ is the learning objective for the GP (Rasmussen and Williams, 2005). $\rho$ denotes the regularization weight; as we demonstrate in Section 6, we initialize the regularization weight $\rho$ to keep the penalty at the same order of magnitude as the negative log likelihood. An alternative training process is to minimize the regression loss and the collision penalty alternatively. We observe in our empirical study that both training processes could lead to reasonable convergence behavior of the LOCO training loss.

## 5 DISCUSSION AND THEORETICAL INSIGHT

This subsection discusses the theoretical insight underlying the collision-free regularizer, by inspecting the effect of regularization on the regret bound of LOCO where the constantly trained neural network feeds a dynamic embedding to the Gaussian process.

While the key idea for bounding the regret of UCB-based GP bandit optimization algorithms follows the analysis of Srinivas et al. (2010), two unique challenges are posed in the analysis of LOCO. Firstly, unlike previous work in Srinivas et al. (2010), the neural network is constantly retrained along with the new observations. Thus, the input space for the downstream Gaussian process could be highly variant.

For the discussion below, we consider a stationary and monotonic kernel, and assume that retraining the neural network $g$ does not decrease the distance between data points in the latent space. It is worth noting that, although not strictly enforced, such monotonicity behavior is naturally encouraged by our proposed regularization, which only penalizes the pair of too-close data in the latent space. Under the above assumption, the internal complexity of neural network training still makes it challenging to bound the regret w.r.t the dynamics of the neural network. Thus, we investigate the dynamics of the mutual information term in the regret bound, and justify the proposed collision-free regularizer by showing that penalizing the collisions tends to reduces the upper bound on the regret.

We first consider a discrete decision set and then leverage the desired Lipschitz continuity on the regularized space to extend our results to the continuous setting (cf full proofs in Appendix A).

**Proposition 1.** *Let $\mathcal{Z}$ be a finite discrete set. Let $\delta \in (0, 1)$, and define $\beta_t = 2\log(|\mathcal{Z}|t^2/6\delta)$. Suppose that the objective function $h : \mathcal{Z} \times \theta \to \mathcal{R}$ defined on $\mathcal{Z}$ and parameterized by $\theta$ is a sample from GP. Furthermore, consider a stationary and monotonic kernel, and assume that retraining the neural network $g$ does not decrease the distance between data points in the latent space. Running GP-UCB with $\beta_t$ for a sample $h$ of a GP with mean function zero and stationary covariance function $k(x, x')$, we obtain a regret bound of $\mathcal{O}^*(\sqrt{\log(|\mathcal{Z}|)T(\gamma_T - \mathbb{I}(h(z_T, \theta_{h,0}); \phi_T))})$ with high probability.*

*More specifically, with $C_1 = 8/\log(1 + \sigma^{-2})$, we have*

$$\mathbb{P}\left[R_T \leq \sqrt{C_1 T \beta_T (\gamma_T - \mathbb{I}(h(z_T, \theta_{h,0}); \phi_T))}\right] \geq 1 - \delta.$$

*Here $\gamma_T$ is the maximum information gain after T iterations, and $\phi_T$ as the identification of the collided data points on $\mathcal{Z}$. $\gamma_T$ is defined as $\gamma_T \coloneqq \max_{A \subset \mathcal{Z}, |A|=T} \mathbb{I}(y_A, \phi_T; h(A, \theta_T))$.*

The collision regularization reduced the maximum mutual information by a specific term dependent on the distribution of the noise caused by the collision of data points. The distribution is dynamic and determined by the complex learning process of the neural network. In the following, we show that the mutual information is bounded within a given interval:

Assume $\phi_t$ is a random variable that identify $z_t \in \mathcal{Z}$, $y_t \in \mathcal{Y}$ from its collided points, and the variance of the collision is $\sigma_{\text{col}}^2$, then we have

$$0 \leq \mathbb{I}(h(z_T, \theta_{h,0}); \phi_T) \leq 1/2 \log |2\pi e \sigma_{\text{col}}^2 I|$$

This means that if $\phi_t$ is a random variable sampled from a Gaussian distribution defined on $h$, then $\mathbb{I}(h(z_T, \theta_{h,0}); \phi_T)$ is maximized.

The regularization also constrains the function $h$ to be Lipschitz-continuous with a Lipschitz constant, enabling a slightly narrower regret bound.

**Proposition 2.** *Let $\mathcal{Z} \subset [0, r]^d$ be compact and convex, $d \in N, r > 0$ and $\lambda \geq 0$. Suppose that the objective function $h : \mathcal{Z} \times \theta \to \mathcal{R}$ defined on $\mathcal{Z}$ and parameterized by $\theta$ is a sample from GP and is Lipschitz continuous with Lipschitz constant $\lambda$. Let $\delta \in (0, 1)$, and define $\beta_t = 2\log(\pi^2 t^2/6\delta) + 2d\log(\lambda r d t^2)$. Furthermore, consider a stationary and monotonic kernel, and assume that retraining the neural network $g$ does not decrease the distance between data points in the latent space. Running GP-UCB with $\beta_t$ for a sample $h$ of a GP with mean function zero and stationary covariance function $k(x, x')$, we obtain a regret bound of $O^*(\sqrt{dT(\gamma_T - \mathbb{I}(h(z_T, \theta_{h,0}); \phi_T))})$ with high probability.*

*More specifically, with $C_1 = 8/\log(1 + \sigma^{-2})$, we have*

$$\mathbb{P}\left[R_T \leq \sqrt{C_1 T \beta_T (\gamma_T - \mathbb{I}(h(z_T, \theta_{h,0}); \phi_T)) + 2}\right] \geq 1 - \delta.$$

*Here $\gamma_T$ is the maximum information gain after T iterations, and $\phi_T$ as the identification of the collided data points on $\mathcal{Z}$. $\gamma_T$ is defined as $\gamma_T \coloneqq \max_{A \subset \mathcal{Z}, |A|=T} \mathbb{I}(y_A, \phi_T; h(A, \theta_T))$.*

## 6 EXPERIMENTS

In this section, we empirically evaluate our algorithm on several synthetic and real-world benchmark blackbox function optimization tasks. All experiments are conducted on Google Cloud GPU instance (4 vCPUs, 15 GB memory, Tesla T4 GPU) and Google CoLab high-RAM GPU instance.

### 6.1 EXPERIMENTAL SETUP

We consider five baselines in our experiments. Three popular optimization algorithms—particle swarm optimization (PSO) (Miranda, 2018), Tree-structured Parzen Estimator Approach (TPE) (Bergstra et al., 2011), a BoTorch (Balandat et al., 2020) implementation of Trust Region Bayesian Optimization (TuRBO) (Eriksson et al., 2019), and standard Bayesian optimization (BO) (Nogueira, 2014) which uses Gaussian processes as the statistical model—are tuned in each task. Another baseline we consider is the sample-efficient LSO (SE LSO) algorithm, which is implemented based on the algorithm proposed by Tripp et al. (2020). We also compare the non-regularized latent space optimization (LSO), LOCO with uniform weights (i.e. $\zeta = 0$, referred to as LOCO), and the dynamically-weighted LOCO (i.e. with $\zeta > 0$, referred to as DW LOCO) proposed in this paper.

One crucial problem in practice is tuning the hyper-parameters. The hyper-parameters for GP are tuned for periodically retraining in the optimization process by minimizing the loss function on a

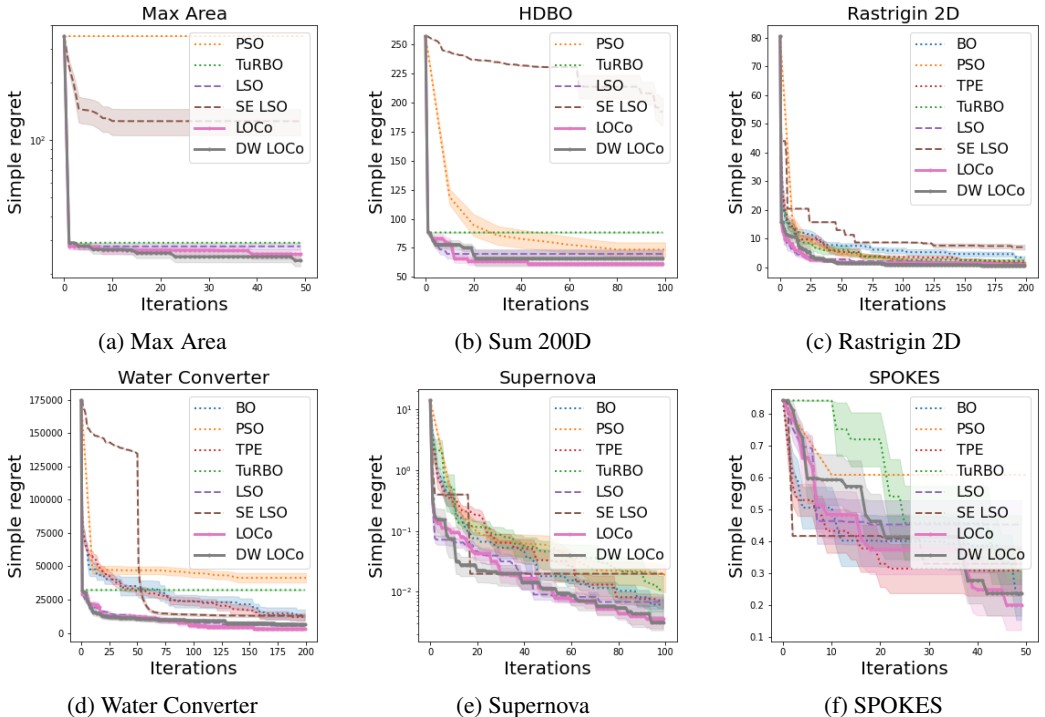

Figure 3: Experiment results on six pre-collected datasets. Each experiment is repeated at least eight times. The colored area around the mean curve denotes the $\frac{\hat{\sigma}}{\sqrt{n}}$. Here $\hat{\sigma}$ denotes the empirical standard deviation. $n$ denotes the number of cases repeated in experiments. Note that the BO and TPE implementation we tested can not finish in reasonable time for figure 3a and figure 3b.

validation set. For all our tasks, we choose a simple neural network architecture due to the reasoning in section 3.3, as well as due to limited and expensive access to labeled data under the BO setting. The coefficient $\rho$ is, in general, selected to guarantee a similar order for the collision penalty to GP loss. The $\lambda$ should be tolerant of the additive noise in the evaluation. In practice, we choose the simple setting $\lambda = 1$ and find it perform well. We also include a study of the parameter choice in the appendix. $\zeta$ controls the aggressiveness of the importance weight. While $\zeta$ should not be too close to zero (equivalent to uniform weight) , an extremely high value could make the regularization overly biased. Such a severe bias could allow a heavily collided representation in most of the latent space and degrade regularization effectiveness. The value choice is similar to the inverse of the temperature parameter of softmax in deep learning (Hinton et al., 2015). Here we use $\zeta = 1$ for simplicity and find it robust to different tasks. All experiments are conducted on the pre-collected datasets. We defer the detailed experimental setup to Appendix C.

## 6.2 DATASETS AND RESULTS

We now evaluate LOCo on three synthetic datasets and three real-world datasets. We demonstrated the improvement in LOCo that is enabled by the explicit collision mitigation in the lower-dimensional latent space in terms of average simple regret.

**2D Shape Area Maximization** The dSprites dataset (Matthey et al., 2017) consists of images of size $64 \times 64$ containing 2d Shapes with different scales, rotations, and positions. Each pixel value of the images are binary, hence $x \in \{0, 1\}^{64 \times 64}$. The goal is to generate a shape $x$ with maximum area, which is equivalent to finding $\arg\max_x \sum_i^{64 \times 64} x_i$ where $i$ corresponds to the pixel index. The neural network is pretrained on 50 data points. To meet the limitation of memory on our computing instance, we uniformly sample 10000 points from the original dataset and approximately maintain the original distribution of the objective value. The DW LOCo and LOCo outperform or matches the baseline methods on this dataset.

**Sum 200D** We create a synthetic dataset Sum 200D of 200 dimensions. Each dimension is independently sampled from a standard normal distribution to maximize the uncertainty on that dimensions and examine the algorithm's capability to solve the medium-dimensional problem. We want to maximize the label $f(x) = \sum_{i=1}^{200} e^{x_i}$ which bears an additive structure and of non-linearity. The neural network is pretrained on 100 data points. As illustrated by figure 3b DW LOCO and LOCO could significantly outperform baselines that do not specifically leverage the additive structures of the problem.

**Rastrigin-2D** The Rastrigin function is a non-convex function used as a performance test problem for optimization algorithms. It was first proposed by Rastrigin (1974) and used as a popular benchmark dataset for evaluating Gaussian process regression algorithms (Cully et al., 2018). Formally, the 2D Rastrigin function is $f(x) = 10d + \sum_{i=1}^{d} x_i^2 - 10\cos(2\pi x_i)$, $d = 2$. For convenience of comparison, we take the $-f(x)$ as the objective value to make the optimization tasks a maximization task.

**Supernova-3D** Our first real-world task is to perform maximum likelihood inference on three cosmological parameters, the Hubble constant $H_0 \in (60, 80)$, the dark matter fraction $\Omega_M \in (0, 1)$, and the dark energy fraction $\Omega_A \in (0, 1)$. The likelihood is given by the Robertson-Walker metric, which requires a one-dimensional numerical integration for each point in the dataset from Davis et al. (2007). The neural network is pretrained on 100 data points. As illustrated by figure 3e, both LOCO and DW LOCO demonstrate its consistent robustness. Among them, DW LOCO slightly outperform LOCO the early stage.

**Water Converter Configuration-16D** This UCI dataset we use consists of positions and absorbed power outputs of wave energy converters (WECs) from the southern coast of Sydney. The applied converter model is a fully submerged three-tether converter called CETO. 16 WECs locations are placed and optimized in a size-constrained environment.

**Redshift Distribution-14D** Careful accounting of all the requirements and features of these experiments becomes increasingly necessary to achieve the goals of a given cosmic survey. SPOKES (SPectrOscopic KEn Simulation) is an end-to-end framework that can simulate all the operations and critical decisions of a cosmic survey (Nord et al., 2016). In this work, we use SPOKES to generate galaxies within a specified window of distances from Earth. We then minimize the Hausdorff distance between the desired redshift distribution and the simulation of specific cosmological surveys generated by SPOKES. In our experiments, the neural network is pretrained with 400 data points. As illustrated by figure 3f, the simple regret of DW LOCO drops slower yet eventually outperforms or matches other baselines' performances.

In general, our experimental results consistently demonstrate the robustness of our methods against collisions in the learned latent space. Our method outperforms or matches the performance of the best baselines in all scenarios. When compared to the sample-efficient LSO, DW LOCO performs better in most cases and shows a steady capability to reach the optimum by explicitly mitigating the collision in the latent space. Due to the dynamics of representation learning process, it is difficult to claim that the performance improvement brought by dynamic weighting is universal. This aligned with the observation in the experiments that DW LOCO brings observable improvement in the regret curve at a certain stage for an optimization task and achieve an ultimate performance that at least matches LOCO. In contrast, the sample-efficient LSO might fail due to the collision problem.

## 7 CONCLUSION

We have proposed a novel regularization scheme for latent-space-based Bayesian optimization. Our algorithm—namely LOCO—addresses the collision problem induced by dimensionality reduction and improves the performance for latent space-based optimization algorithms. We show that the regularization effectively mitigates the collision problem in the learned latent spaces and, therefore, can boost the performance of the Bayesian optimization in the latent space. We demonstrate solid empirical results for LOCO on several synthetic and real-world datasets. Furthermore, we demonstrate that LOCO can deal with high-dimensional input that could be highly valuable for real-world experiment design tasks such as cosmological survey scheduling.

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

# A PROOFS

In this section, we provide proofs for our main theoretical results (Proposition 1 and Proposition 2).

## A.1 PROOF OF PROPOSITION 1: REGRET BOUND ON DISCRETE DECISION SET

We follow the proof structure in Srinivas et al. (2010) and introduce new notations to characterize the learning process of the neural network and the collision in the proof.

Before proving Proposition 1, we first introduce a few useful lemmas.

**Lemma 1.** *Pick $\delta \in (0,1)$ and set $\beta_t = 2\log(|D|\pi_t/\delta)$, where $\sum_{t\geq 1}\pi_t^{-1} = 1$, $\pi_t > 0$. Then with probability $\geq 1 - \delta$, $\forall x \in D, \forall t \geq 1$*

$$|h(g(x,\theta_{g,t-1}),\theta_{t-1}) - \mu_{t_1}| \leq \beta_t^{1/2}\sigma_{t-1}(g(x,\theta_{g,t-1}))$$

*Here $\theta_{g,t-1}$ is the parameter for $g$ at time step $t-1$. $\theta_{h,t-1}$ is the parameter for $h$ at time step $t-1$.*

*Proof.* $\forall x \in D, \forall t \geq 1$, Conditioned on $\mathbf{y}_{t-1} = (y_1,...,y_{t-1})$, $\{x_1,...,x_{t-1}\}$ are deterministic, and $h(g(x,\theta_{g,t-1}),\theta_{h,t-1}) \sim N(\mu_{t-1}(g(x,\theta_{g,t-1})),\sigma_{t-1}^2(g(x,\theta_{g,t-1})))$. Then using the subgaussianity of $h$, we have

$$Pr\{|h(g(x,\theta_{g,t-1}),\theta_{t-1}) - \mu_{t_1}| \geq \beta_t^{1/2}\sigma_{t-1}(g(x,\theta_{g,t-1}))\} \leq e^{-\beta_t/2}$$

Applying the union bound, with probability $\geq 1 - |D|e^{-\beta_t/2}$, $\forall x \in D$

$$|h(g(x,\theta_{g,t-1}),\theta_{t-1}) - \mu_{t_1}| \leq \beta_t^{1/2}\sigma_{t-1}(g(x,\theta_{g,t-1}))$$

Let $|D|e^{-\beta_t/2} = \delta/\pi_t$, applying the union bound for $\forall t \in \mathbb{N}$ the statement holds. $\qquad\square$

**Lemma 2.** *Consider a stationary and monotonic kernel, and assume that retraining the neural network $g$ does not decrease the distance between data points in the latent space. Then $\forall t \geq 1$,*

$$r_t(\theta_{t-1}) \leq 2\beta_t^{1/2}\sigma_{t-1}(g(x,\theta_{g,t-1})) \leq 2\beta_t^{1/2}\sigma_{t-1}(g(x,\theta_{g,T})).$$

*Here $\theta_{g,T}$ is the parameter for $g$ at time step $T$.*

*Proof.*

$$\begin{aligned}
r_t(\theta_{t-1}) &= h(g(x^*,\theta_{g,t-1}),\theta_{h,t-1}) - h(g(x_t,\theta_{g,t-1}),\theta_{h,t-1}) \\
&\leq \beta_t^{1/2}\sigma_{t-1}(g(x^*,\theta_{g,t-1})) + \mu_{t-1}(g(x^*,\theta_{g,t-1}) - h(g(x_t,\theta_{g,t-1}),\theta_{h,t-1}) \\
&\leq \beta_t^{1/2}\sigma_{t-1}(g(x_t,\theta_{g,t-1})) + \mu_{t-1}(g(x_t,\theta_{g,t-1}) - h(g(x_t,\theta_{g,t-1}),\theta_{h,t-1}) \\
&\leq 2\beta_t^{1/2}\sigma_{t-1}(g(x_t,\theta_{g,t-1})) \\
&\leq 2\beta_t^{1/2}\sigma_{t-1}(g(x_t,\theta_{g,T}))
\end{aligned}$$

The last line is because the non-decreasing distance between $g(x,\theta_{g,T})$ and $g(x',\theta_{g,T})$ $\forall x, x' \in D$ leads to larger variance $\sigma_t$ when using a stationary and monotonic kernel. $\qquad\square$

**Lemma 3.** *The information gain for the points selected can be expressed in terms of the predictive variance. If $\mathbf{h}_T = (h(g(x_t,\theta_{g,T}),\theta_{h,T})) \in \mathbb{R}^T$:*

$$\mathbb{I}(\mathbf{y}_T;\mathbf{h}_T) = \frac{1}{2}\sum_{t=1}^{T}\log(1 + \sigma^{-2}\sigma_{t-1}^2(g(x_t,\theta_{g,T})))$$

*Proof.* First, we get $\mathbb{I}(\mathbf{y}_T; \mathbf{h}_T) = H(\mathbf{y}_T) - \frac{1}{2} \log |2\pi e \sigma^2 \mathbf{I}|$. Then,

$$H(\mathbf{y}_T) = H(\mathbf{y}_{T-1}) + H(y_T | \mathbf{y}_{T-1})$$
$$= H(\mathbf{y}_{T-1}) + \log\big(2\pi e(\sigma_{t-1}^2(g(\mathbf{x}_T, \theta_{g,T})))\big)$$

Since $x_1, ..., x_T$ are deterministic conditioned on $\mathbf{y}_{T-1}$. The result follows by induction. □

**Lemma 4.** *The gap of the mutual information between collision-free $g(x_t, \theta_{g,T})$ and unregularized $g(x_t, \theta_{g,0})$ is*

$$\mathbb{I}(\mathbf{y}_T; h(z_T, \theta_{h,T})) = \mathbb{I}(\mathbf{y}_T; h(z_T, \theta_{h,T})|\phi_T)$$
$$= \mathbb{I}(\mathbf{y}_T; h(z_T, \theta_{h,0})|\phi_T)$$
$$= \mathbb{I}(\mathbf{y}_T, \phi_T; h(z_T, \theta_{h,0})) - \mathbb{I}(h(z_T, \theta_{h,0}); \phi_T))$$

*Here $z_t = g(x_t, \theta_{g,T})$. $\phi$ is the identification of collided data points.*

The result is a simple application of the chain rule of mutual information. $\mathbb{I}(\mathbf{y}_T; \mathbf{h}_T) = \mathbb{I}(\mathbf{y}_T; h(z_T, \theta_{h,T}))$ corresponds to the information gain under fully regularized and assumed collision-free setting. $\mathbb{I}(\mathbf{y}_T, \phi_T; h(z_T, \theta_{h,0}))$ corresponds to information gain under unregularized setting.

**Lemma 5.** *Pick $\delta \in (0, 1)$ and let $\beta_t$ be defined as in Lemma 1. Then, the following holds with probability $\geq 1 - \delta$, $\forall T \geq 1$,*

$$\sum_{t=1}^{T} r_t^2 \leq \beta_T C_1 \mathbb{I}(\mathbf{y}_T; \mathbf{h}_T) \leq C_1 \beta_T (\gamma_T - \mathbb{I}(h(z_T, \theta_{h,0}); \phi_T)).$$

*Here $C_1 := \frac{8}{\log(1+\sigma^{-2})} \geq 8\sigma^2$.*

*Proof.* We first observe that

$$4\beta_t \sigma_{t-1}^2(g(x_t, \theta_{g,T})) \leq 4\beta_t \sigma^2(\sigma^{-2}\sigma_{t-1}^2(g(x_t, \theta_{g,T})))$$
$$\leq 4\beta_t \sigma^2\big(\frac{\sigma^{-2}}{\log(1+\sigma^{-2})}\big) \log\big(1 + \sigma^{-2}\sigma_{t-1}^2(g(x_t, \theta_{g,T}))\big)$$

Combining the above inequality with Lemma 2, Lemma 3 and Lemma 4 completes the proof. □

Now we are ready to prove Proposition 1.

*Proof of Proposition 1.* Proposition 1 is a simple consequence of Lemma 4 and Lemma 5 and Cauchy-Schwarz inequality. □

## A.2 PROOF OF PROPOSITION 2: REGRET BOUND WITH LIPSCHITZ-CONTINUOUS OBJECTIVE FUNCTION

We first modify Lemma 5.7 and Lemma 5.8 in Srinivas et al. (2010) since we are assuming the deterministic Lipschitz-continuity for $h$. Use the same analysis tool $Z_t$ defined as a set of discretization $Z_t \subset Z$ where $Z_t$ will be used at time $t$ in the analysis.

We choose a discretization $Z_t$ of size $(\tau_t)^d$. so that $\forall z \in Z$,

$$||z - [z]_t||_1 \leq rd/\tau_t \tag{4}$$

where $[z]_t$ denotes the closest point in $Z_t$ to $z$.

**Lemma 6.** *Pick $\delta \in (0, 1)$ and set $\beta = 2\log(\pi_t \delta) + 2d \log(Lrdt^2)$, where $\sum_{t \geq 1} \pi_t^{-1} = 1$, $\pi_t > 0$. Let $\tau_t = Lrdt^2$. Hence then*

$$|h(z^*, \theta_{h,t-1}) - \mu_{t-1}([z^*]_t)| \leq \beta_t^{1/2} \sigma_{t-1}([z^*]_t) + 1/t^2 \quad \forall t \geq 1$$

*holds with probability $\geq 1 - \delta$.*

*Proof.* Using the Lipschitz-continuity and equation 4, we have that

$$\forall z \in Z, |h(z, \theta_{h,t-1}) - h([z]_t, , \theta_{h,t-1})| \leq Lrd/\tau_t$$

By choosing $\tau_t = Lrdt^2$, we have $|Z_t| = (Lrdt^2)^d$ and

$$\forall z \in Z, |h(z, \theta_{h,t-1}) - h([z]_t, \theta_{h,t-1})| \leq 1/t^2$$

Then using Lemma 1, we reach the expected result. □

Based on Lemma 2 and Lemma 6, we could have the following result directly.

**Lemma 7.** *Pick $\delta \in (0,1)$ and set $\beta = 2\log(2\pi_t\delta) + 2d\log(Lrdt^2)$, where $\sum_{t\geq 1}\pi_t^{-1} = 1$, $\pi_t > 0$.*
*Then with probability $\geq 1 - \delta$, for all $t \in N$, the regret is bounded as follows:*

$$r_t \leq 2\beta_t^{1/2}\sigma_{t-1}(z_t) + 1/t^2$$

*Proof.* Using the union bound of $\delta/2$ in both Lemma 2 and Lemma 6, we have that with probability $1 - \delta$:

$$\begin{aligned}
r_t &= h(z^*) - h(z_t) \\
&\leq \beta_t^{1/2}\sigma_{t-1}(z_t) + 1/t^2 + \mu_{t-1}(z_t) - h(z_t) \\
&\leq 2\beta_t^{1/2}\sigma_{t-1}(z_t) + 1/t^2
\end{aligned}$$

which complete the proof. □

Now we are ready to prove Proposition 2.

*Proof of Proposition 2.* Using Lemma 7, we have that with probability $\geq 1 - \delta$:

$$\sum_{t=1}^{T} 4\beta_t\sigma_{t-1}^2(x_t) \leq C_1\beta_T(\gamma_T - \mathbb{I}(h(z_T, \theta_{h,0}); \phi_T)) \quad \forall T \geq 1$$

By Cauchy-Schwarz:

$$\sum_{t=1}^{T} 2\beta_t^{1/2}\sigma_{t-1}(x_t) \leq \sqrt{C_1\beta_T(\gamma_T - \mathbb{I}(h(z_T, \theta_{h,0}); \phi_T))} \quad \forall T \geq 1$$

Finally, substitute $\pi_t$ with $\pi^2 t^2/6$ (since $\sum 1/t^2 = \pi^2/6$). Proposition 2 follows. □

## B    DEMONSTRATION OF THE COLLISION EFFECT

### B.1    VISUALIZATION OF THE COLLISION EFFECT IN THE LATENT SPACE

We demonstrate the collision effect in the latent space. We train the same neural network on Feynman dataset with 101 data points which demonstrate the latent space after two retrains with the retrain interval set to be 50 data points. The regularized one employs DW LOCO, with the regularization parameter $\rho = 1e^5$, penalty parameter $\lambda = 1e^{-2}$, retrain interval $\tilde{T}$, weighting parameter $\gamma = 1e^{-2}$ and the base kernel set to be square exponential kernel. The non-regularized one employs LSO.

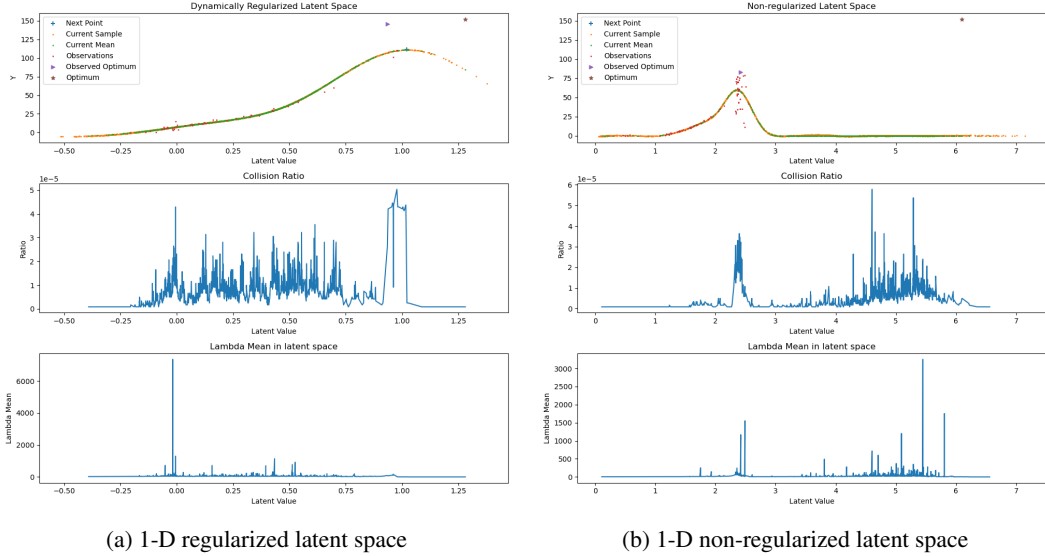

(a) 1-D regularized latent space          (b) 1-D non-regularized latent space

Figure 5: Illustrate the 1-D latent space of Feynman III.9.52 dataset. The second row shows the ratio that the penalty define as equation 1 is non-zero. The third row shows point-wise estimation of $\lambda$. 5a shows a regularized latent space with a few observable collisions. 5b shows a non-regularized latent space with bumps of collisions especially around the maxima among the observed data points. Besides, having fewer collisions in the latent space contribute to the optimization through improving the learned Gaussian process. We observe in this comparison that the next point selected by the acquisition function of the regularized version is approaching the global optima, while the next point in the non-regularized version is trying to solve the uncertainty brought by the severe collision near the currently observed maxima.

### B.2    THE COLLISION EFFECT ON PROPER NEURAL NETWORKS

In this section, we provide empirical results supporting the claim in section 3.3 that increasing the network complexity often does *not* help to reduce the collision in the latent space.

For Feynman task, we test the single-layer neural network, which consists of 10, 1001, 5000, or 6000 neurons with Leaky Relu activation functions.

For Max Area task, we test the three-layer dense neural network. The first layer consists of 50, 100, 1000 or 1500 neurons with Tanh activation functions. The second layer consists of 50 neurons with Tanh activation functions. The third layer consists of 10 neurons with Leaky Relu activation functions.

For Rastrigin-2D task, we test the single-layer neural network, which consists of 10, 100, 1000, or 5000 neurons with Leaky Relu activation functions.

For SPOKES task, we test the single-layer neural network, which consists of 10, 100, 1000, or 2000 neurons with Leaky Relu activation functions.

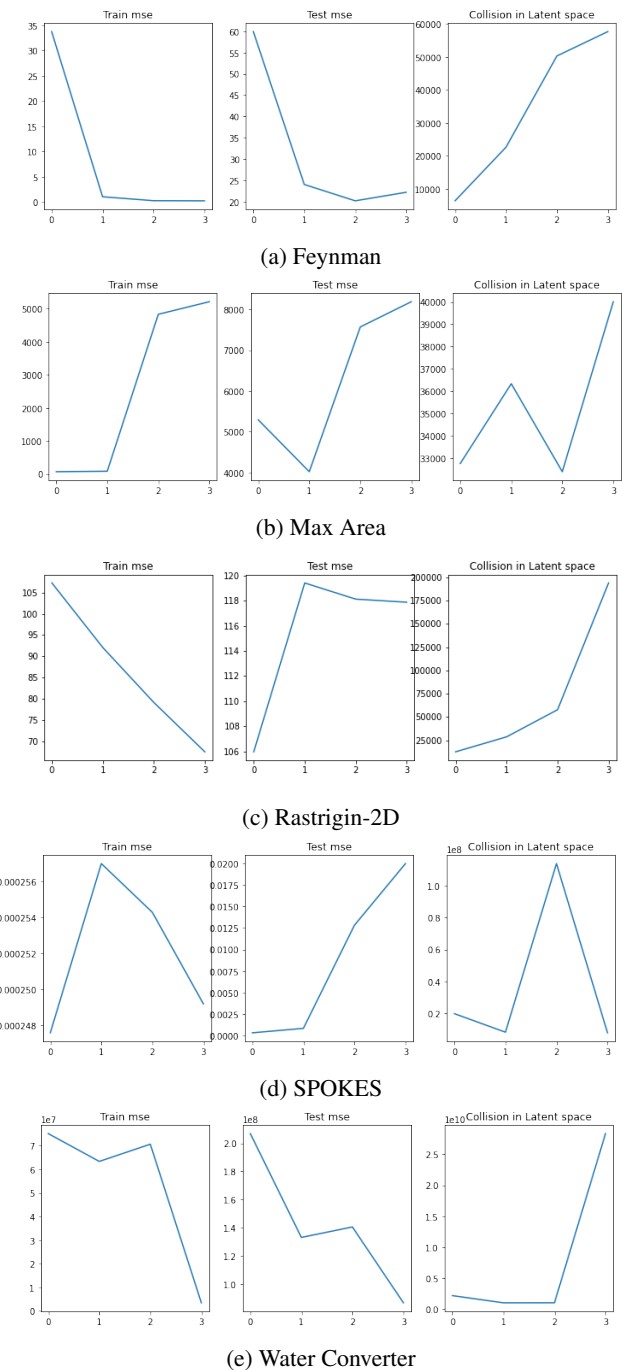

Figure 6: These curves show the network design test results. The collision value shown here is the penalty term proposed in equation 1. The x-axis denotes the neural network's general complexity. The collisions for model with lowest test MSE are still significant.

For the Water Converter task, we test the three-layer dense neural network. The first layer consists of 512, 1024, 2048, or 4096 neurons with Tanh activation functions. The second layer consists of half of the first layer's neurons with Tanh activation functions. The third layer consists of half of the second layer's neurons with Leaky Relu activation functions.

We demonstrate the collision effect on regression task on Rastrigin-2D.

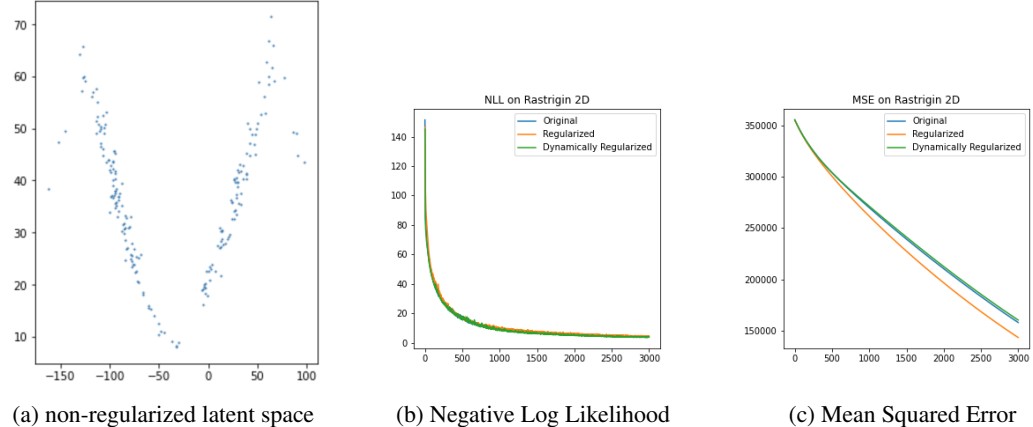

(a) non-regularized latent space      (b) Negative Log Likelihood      (c) Mean Squared Error

Figure 7: Illustrate regression task on Rastrigin-2D dataset. 7a shows a non-regularized latent space after sufficiently trained as is demonstrated in 7b. 7b shows the NLL of the training process. 7a shows the corresponding MSE.

As illustrated in figure 7a, even after being sufficiently trained after 3000 iterations, there is still collisions in the latent space especially around the optima. figure 7a shows that by regularizing the latent space, the ultimate MSE could also be improved.

## C  SUPPLEMENTAL MATERIALS ON ALGORITHMIC DETAILS

Our implementation of LOCO and DW LOCO is built upon the open source package GPytorch (Gardner et al., 2018). The deep kernel is trained with back propogation. We use the Adam (Kingma and Ba, 2014) optimizer with learning rate set to be $1e^{-2}$. Below we discuss the detailed configuration of the underlying neural network and the choice of the key parameters used by the algorithm.

### C.1  ALGORITHMIC DETAILS ON NEURAL NETWORK ARCHITECTURE

As the primary goal of our paper was to showcase the performance of a novel collision-free regularizer, we pick our network architectures to be basic multi-layer dense neural network. We use a 4-layer dense neural network. Its hidden layers consist of 1000, 500, 50 neurons respectively, each with Leaky Relu activation functions. The output layer also uses Leaky Relu as its activation function and generates a 1-dimensional output.

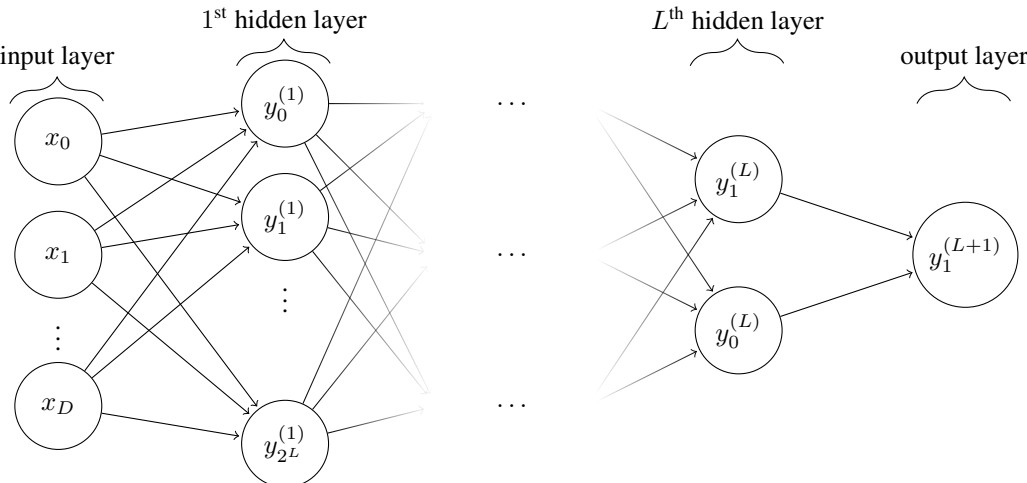

Figure 8: Network graph of a $(L+1)$-layer dense network with $D$ input units and 1 output units. In our experiments, L is set to be 3.

**Pre-training of the Neural Network**    We randomly select the points from the bottom of the dataset to avoid the pre-training simplifying the optimization task while helping with initializing the parameters of the model. The points are used to initialize the neural network instead of being served as initial selections of the optimization task. Without the pre-training stage, the latent embedding fed by the neural network to the gaussian process would be random. The practical problem with such randomness could be a much larger variance for the results since it influences the following neural network training process and the optimization process. In practice, it's always possible to collect pre-training datasets from relative domains when aiming at optimizing unknown objective functions. This pre-training method was also reported in the literature (Snoek et al., 2015).

## C.2 PARAMETER CHOICES

We investigate the robustness of parameter choices of the regularization parameter $\lambda$ on the Rastrigin 2D dataset. We show the results in the figure below.

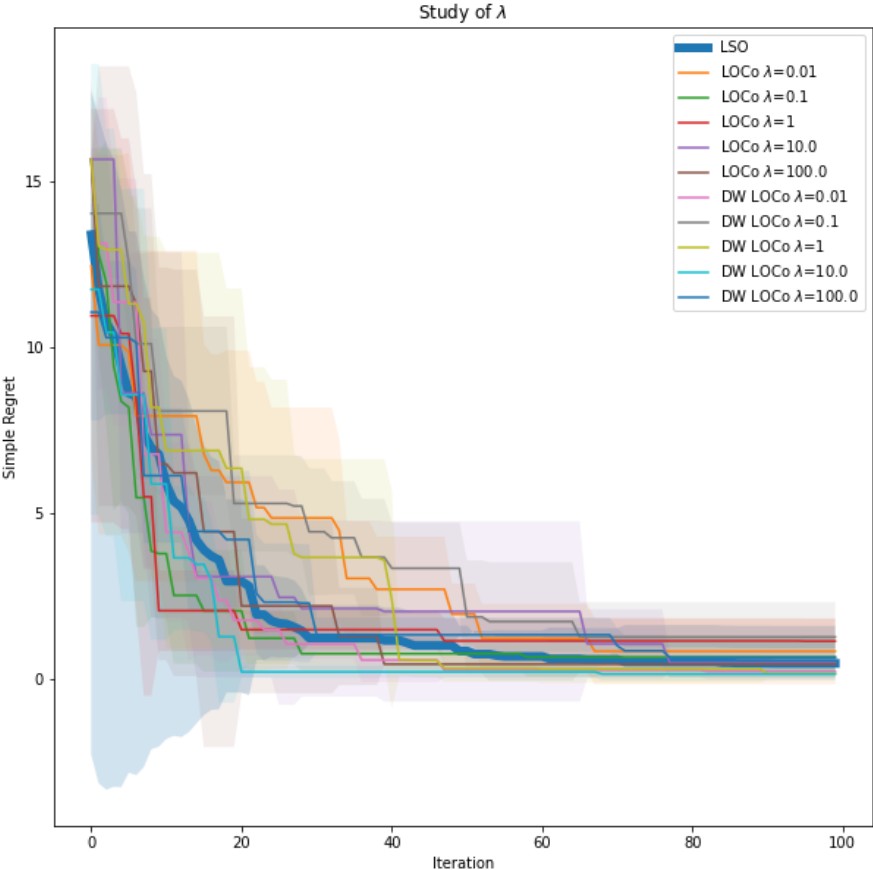

Figure 9: Simple regret under different parameter settings on the Rastrigin 2D dataset. The colored area represents the standard error of the tests at certain iteration. Each experiments are repeated eight times. The figure shows that a moderately large $\lambda$ suffices to achieve decent performance in terms of simple regret. We believe that the wide range of objective values of the test dataset, which otherwise would hurt the optimization performance, can be regularized by the collision penalty. The curves demonstrate the decent performance of DW LOCO as long as the parameters are not set to be too small.

# D ADDITIONAL RESULTS

*Random EMbedding Bayesian Optimization* (REMBO) (Wang et al., 2016) leverages simple random linear transformations to improve the efficiency in low-effective-dimension high-dimensional tasks. We compare LOCO and DW LOCO with the performance of this random-embedding-based method and empirically exposed the failure case of REMBO when its modeling assumption does not hold (i.e. when dealing with dataset that has large effective dimensions).

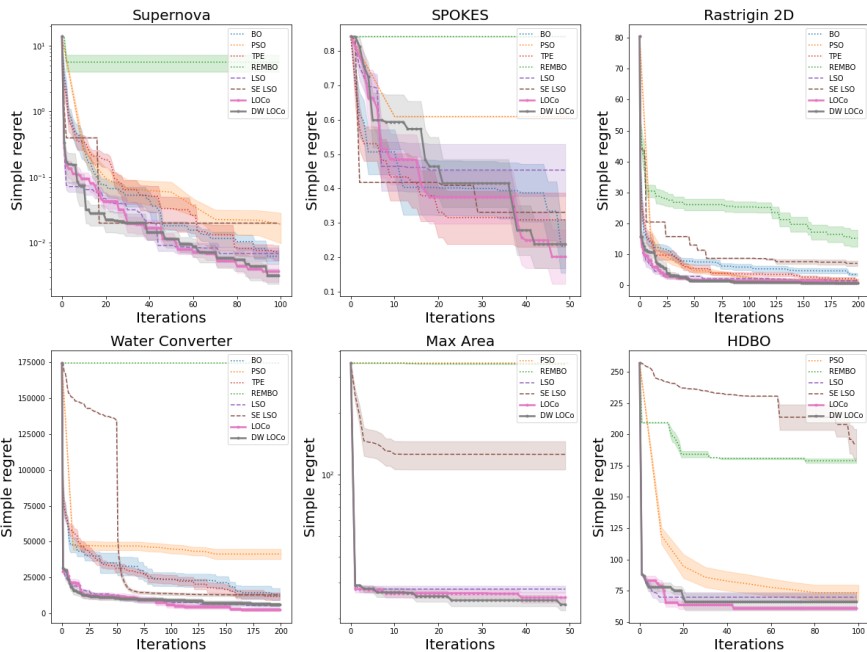

Figure 10: Experiment results on six synthetic & real datasets. Each experiment is repeated at least eight times. The shaded area around the mean curve denotes the $\frac{\hat{\sigma}}{\sqrt{n}}$. Here $\hat{\sigma}$ denotes the empirical standard deviation. $n$ denotes the number of cases repeated in experiments. As illustrated in the figure, the random-embedding-based methods have been significantly outperformed by LOCO and DW LOCO. We place the discussion over REMBO here for two reasons. Firstly, there has been several problems about REMBO as discussed in section 2. Secondly, the experiments are conducted on tasks where the effective dimensions are at a similar scale as the dimensionality of the original inputs and doesn't align with the assumption of REMBO.

