# OpenReview forum: "Learning Representation for Bayesian Optimization with Collision-free Regularization"
_ICLR.cc/2022/Conference — ICLR 2022 Submitted_

### Official Review · Reviewer_G6dp · 2021-10-20

**Correctness:** 4
**Technical Novelty And Significance:** 2
**Empirical Novelty And Significance:** 2
**Recommendation:** 5
**Confidence:** 3

**Main Review:**

- Strengths
1. The paper addresses the problematic behavior (collision) that it deserves and that has not been explicitly discussed well and proposes a solution to the collision
2. The paper adds a proof technique which may be useful for future GP bandit methods using evolving data points.
3. The paper is well written so that it is easy to follow.
- Weaknesses
1. By introducing the collision penalty, the loss function has more hyper-parameters which unavoidably calls for some heuristic. Many hyper-parameters including neural network architectures are different between BO experiments and collision ablation study in the appendix. For the same hyperparameters used in BO exp, is it consistently observed that the collision is well suppressed?
2. With the added collision penalty, GP kernel hyper-parameters cannot be optimized contrast to ordinary GP training. Would it be possible to frame this approach in more Bayesian way (e.g interpreting collision penalty as some sort of prior on network parameter structure) so that more principled approach to handle newly introduced hyper-parameters?
3. “Dynamical weighting” does not seem to make noticeable differences empirically, any formal argument to support this?
- Minor points
1. “GP are not intrinsically designed to deal with structured input” sounds a bit controversial, any reference for this?

**Summary Of The Paper:**

Focusing on Bayesian optimization using latent spaces to overcome the difficulties coming from the high dimensionality of the search space, the paper proposes identifies a problematic behavior called the collision, and proposes a regularizer to mitigate the collision and in turn to enhance the BO performance.

**Summary Of The Review:**

The paper focuses on an issue in latent space based Bayesian optimization and proposed a solution for it. It also appears to provide a useful theoretical tool in regret analysis. However, the added complexity in the pair-wise loss and how new hyperparameters are handled deserves more through theoretical or more extensive empirical support. Therefore, I think an improvement is required for the paper’s acceptance. I will be expecting that some of my concerns are answered in the discussion period.

---

### Official Review · Reviewer_aX8w · 2021-10-29

**Correctness:** 2
**Technical Novelty And Significance:** 2
**Empirical Novelty And Significance:** 2
**Recommendation:** 3
**Confidence:** 4

**Main Review:**

MAJOR COMMENTS:

Contribution I:

The paper claims to "expose the limitations of existing latent space optimization approaches due to the collision effect on the latent space." I do not think the paper provides sufficient evidence to support this contribution. Sections 3 and 4 state that collisions are an issue, but nowhere does the paper _demonstrate_ that it is an issue. Fig. 4 in the appendix shows that there are collisions happening, but the paper does not provide any evidence that collisions are detrimental for Bayesian optimization. Simply raising awareness that collisions can happen is not a contribution because that has already been done (see Moroconi et al., 2020). The contribution would be to show that this actually limiting performance of current methods and that was not shown. In Figure 3, as best as I can tell, LSO is the same as LOCo but without the regularization. The performance of LSO on the benchmarks does not appear to be significantly worse than that of LOCo, so it appears that collisions are not a primary limitation of existing latent space optimization approaches.

Another issue related to motivation and the necessity of the methods introduced in the paper is that it isn't clear to me why end-to-end learning (described in the Introduction) isn't sufficient to avoid the most serious collisions that would be harm GP modeling. Consider the two latent spaces in Fig. 7 in the Appendix. Could the authors report the GP marginal log-likelihood for each of them? It looks clear to me that the latent space for the regularized latent space will have much higher log-likelihood than the non-regularized latent space, because having a big high-variance area is going to be bad for likelihood of a model that assumes homoskedastic noise. If the marginal log-likelihood is higher for the regularized latent space as I suspect, then why was the regularization necessary at all? If I fit the latent space end-to-end (i.e., to maximize marginal log likelihood of the GP) then why doesn't that  smooth out the collisions to better match the homoskedastic noise assumption? This is my main concern with the paper: I acknowledge that collisions could be an issue, but I think that end-to-end training is sufficient to resolve the issue, and so I'm not convinced (and did not see evidence in the paper) that the regularization is actually necessary.

Contribution IV:

There are three vital issues with the empirical evaluation, any one of which would be sufficient for me to recommend revision of the paper.

(1) Only one HDBO baseline is included in the comparison (SE LSO). In a field with so much recent development as HDBO I do not think this is appropriate. At the very least I would insist on adding TuRBO (Eriksson et al., 2019) to the comparison since it has a widely available implementation (https://github.com/uber-research/TuRBO) and shows state-of-the-art performance for HDBO problems of the type used in this paper.

(2) Figures show 1 standard error. They should show 2, so that they provide a 95% confidence interval. As is, the error bars are showing a 68% CI for the means, which is both highly non-standard, and also makes it hard to assess which of these differences are significant. It looks like there are some significant differences, but most differences are not significant at a 95% level.

(3) Section 6.2 describes that the neural networks were pre-trained with 100 or 400 data points. How were these chosen? More importantly, were all methods given 100 or 400 extra data points for initialization? In particular, was BO given 100 or 400 points for its initialization so that all points get the same amount of data? I worry that it was not because the text specifies neural networks as getting the additional data. If not, it would be inappropriate to compare the methods by total number of iterations when some methods were allowed more data than others. So please verify that all methods were given the same total amount of data. For instance, when TuRBO is added as an additional baseline, it will be important to set n_init as the same 100 or 400 used as pre-training for the neural network-based methods.


OTHER COMMENTS:

* Missing prior work: Moroconi et al. (2020) studies the issue of latent space collisions for linear embeddings and should be referenced.

* For (1), this is not scale invariant on z. If I multiply z_i and z_j each by C, then I will require a larger value of \lambda to achieve the same collision penalty. The paper proposes setting \lambda using the original X. I don't think that can reasonably be done when the penalty is not scale invariant in Z. I can keep X the same as it is (so lambda stays fixed) and then by adjusting the weights in the neural net could linearly expand Z in a way that doesn't affect the GP, but would drastically change p_ij by changing the scale of Z relative to \lambda. I think this is problematic for the definition of the penalty (it should be scale invariant to the embedding), and also based on this fact I disagree with the claim that \lambda could be set other than through hyperparameter tuning.

* The chosen setting of \lambda = 1 is entirely unjustified and would depend on the scale of f at the very least.

* Unclear notation: Section 3.3 defines the collision property in terms of |g(x_i) - g(x_j)|. g(x_i) is a point in the presumably multi-dimensional latent space, so it is unclear what norm is being used here. Was the latent space always 1-d in this paper? It isn't in others. Same issue in (1).

* Fig. 1: I don't know what "Current Sample" and "Current Mean" are. There is no y axis label.

* theta_h and theta_g are not defined in the paper. I think that theta_h are the GP parameters and theta_g are the NN parameters, but please define.

* Algo 1 Line 3: Shouldn't alpha also be a function of theta_h if those are the GP parameters? If not then what is theta_h?

* Algo 1 Line 5 says that the theta_t, which I understand to be both the entire neural network and the GP hyperparameters, are trained every iteration. But Section 6.1 says "The hyper-parameters for GP are tuned for periodically retraining in the optimization process." Does periodically mean every iteration? If not what is being retrained in Algo 1 Line 5?

* The paper needs editing for grammar.


References:

Moroconi R, Kumar KSS, Deisenroth MP, High-dimensional Bayesian optimization with projections using quantile Gaussian processes, Optimization Letters, 2020

Eriksson D, Pearce M, Gardner J, Turner R, Poloczek M, Scalable global optimization via local Bayesian optimization, NeurIPS, 2019

**Summary Of The Paper:**

The paper studies the use of embeddings for high-dimensional Bayesian optimization and introduces a regularization term into the model training to ensure smoothness in the latent space, so that it will be suitable for GP modeling and Bayesian optimization.

**Summary Of The Review:**

The idea of regularization in the embedding is interesting but I feel is underdeveloped and I do not find that the paper provides sufficient evidence for its claims that the method is necessary or useful. As such I do not think it is ready for publication at this time.

---

### Official Review · Reviewer_DSCs · 2021-11-02

**Correctness:** 3
**Technical Novelty And Significance:** 2
**Empirical Novelty And Significance:** 2
**Recommendation:** 3
**Confidence:** 5

**Main Review:**

I have several questions that I would like to kindly ask the authors, hoping this can improve my assessment of the paper:

1. What is the relation of this work with the recently proposed method of combining metric learning with latent space Bayesian optimisation (Grosnit et. al 2021). At first sight, the problems seem equivalent yet no comparison is made. Can the authors please clarify the main differences in that work?
2. Concerning the theoretical results, I am a bit confused about the novelty of the proof. Does it seem to be a direct application of Srinivas 2010? I do understand the changing neural network idea, however, resolving this by an assumption of monotonic increase seems to answer the difficulty of the proof with an assumption. I would be really grateful if the authors help me understand the challenges and novelty in the proof. Additionally, when presenting the proof, it would be great if there is a section listing all assumptions made. I found it extremely disturbing to have to read the proof not knowing the assumptions made up-front. Please try to improve the proof's exposition.
3. I am a bit confused about the stochastic treatment of the approach. The authors seem to add noise everywhere but do not study its effects in, for example, equation 1. Like equation 1 is defined over observations of y and z and not in expectation? Why would that be a good idea and how does this affect the variance of p_ij?
4. Can the authors please help me understand the relationship between lambda and the Lipschitz constant of the function? If they are the same, how is it possible to estimate that? Estimating the Lipschitz constant is an extremely hard task to do. Additionally, if we are enforcing the net to be Lipshitz, there is a large body of work that is ignored in this paper that should be referred to. In fact, that work leads us to understand that difficulty beyond MLPs to more sophisticated networks. Was this the reason the authors chose simple networks to handle?
5. Arriving at Equation 3 is a bit unprincipled from my point of view. I would appreciate a rigorous derivation from probabilistic assumptions of GPs. What kernels would those amount to and how can Equation 3 be derived from GP log-marginals?
6. Reading the experiments section, I found it very hard to understand. I also tried to find the additional details and descriptions in the appendix but could only find ablation studies. It would be really useful if the authors improve the exposition of this section clearly stating the dimensions of each of the considered tasks. The 2D shape area maximisation is a Toy domain and so is the sum 200D. In the sum 200 D the black-box clearly exhibits an affinity structure to which specific GPs with affine kernels have been developed. I urge the authors to compare to one of those. In the supernova and redshift what are the dimensions of the problem? Are those high-dimensional tasks? I think the authors should run on molecule domains which were one of the motivations of the work in Tripp et. al. 2020. They should also reference and compare to a wide literature of work that handles high-dimensional discrete structures (e.g., Junction Tree VAEs and many others).
6. Figure 1 is unclear: what are the original space points and what are the latent points here? If the star y-value is the true optimum and its x-value is the optimiser, it is true that it is hard to model for the GP as the region where the optimiser is located is surrounded by low y-value points. But I don’t see the point of that Figure, it’s just a panel about how it can be difficult for a GP to model such data. It would have been better to have a real example maybe.
7. What does it mean to minimise uniformly the collisions? Surely the regulariser will be proportional to the difference in function values.
8. Is the GP training jointly with the neural network? Can you please clarify this? If so, what is the relation to GPLVMs?
9. It is said that it is more efficient to learn how to eliminate collisions in regions around the optimiser in the latent space due to limited resources (i.e. labels) in BO. But the dynamic method does not solve the data requirement problem. It merely weighs the penalties based on function values, which is a good first step. I agree that this weighting enables the regulariser to penalise more aggressively collided pairs with high function values. But to compute the weights you nevertheless need all points to have labels. Can you please clarify this point?
10. In the appendix are these architectures really adapted? If I understand your latent space is always 1-dimensional. Isn’t that limiting? Isn’t that actually creating collisions?

Minor:

I think the K_SE in section 3.2 is missing a square inside the exponential.
The paper is really hard to read and understand. I would urge the authors to: 1) write a clear set of assumptions during the proof, 2) clear listings of dimensionalities and structures of networks used in the experiments, and 3) Improve the quality of the Figures using vectorised graphics.
The authors could also cite these two papers: https://arxiv.org/pdf/1812.02833.pdf and https://openaccess.thecvf.com/content_CVPR_2020/papers/Ding_Guided_Variational_Autoencoder_for_Disentanglement_Learning_CVPR_2020_paper.pdf

**Summary Of The Paper:**

This paper is concerned with latent space Bayesian optimisation that typically involves a step of learning a lower-dimensional latent representation. The authors focus on non-linear embeddings generated through neural networks. They observe a collision problem in the latent space and attempt to resolve it by introducing a regulariser based on Lipschitz continuity. In a set of experiments, they demonstrate that such a method is effective in various benchmarks. Although interesting, I still find this paper lacking as presented in the next section.

**Summary Of The Review:**

See above. This paper seems interesting but is not convincing enough for an acceptance.

---

### Official Review · Reviewer_Knwx · 2021-11-04

**Correctness:** 3
**Technical Novelty And Significance:** 3
**Empirical Novelty And Significance:** 2
**Recommendation:** 5
**Confidence:** 4

**Main Review:**


## Merits

- Adapting Bayesian optimization for high dimensional search spaces is arguable one of the key challenges at the moment, and learning embeddings is one of the most promising directions. The paper addresses an important practical problem of these embeddings

- Overall I found the paper to be well written and easy to follow.

- The paper convincingly motivates the occurrence of collisions and the proposed penalty terms seems sound.

- While the empirical evaluation exhibit some flaws (see below), it seems that the proposed approach indeed improves upon baselines.


## Concerns

- Figure 1 is confusing to me. What is the original function? What means current sample? It would be great to visualize the true objective to see how far off the GP prediction is?

- Last sentence in Section 4.2: How exactly do you select \lambda in the input space? Shouldn't lambda be proportional to epsilon (i.e the observation noise)?

- Do you adapt rho during optimization, since the order of magintude of the likelihood might change over time?

- How do you select the pretrained datapoints to train the neural network?

- I am not sure what you mean with "... to meet the limitation of memory on our computing instance, we uniformly sample 10000 points from the original dataset ". This doesn't seem to be a lot of data? How many datapoints does the original benchmark have?

- Why is TPE and BO missing for the Max Area and the HDBO problem?

- It seems that the weighted version of LOCo does not lead to any gain in performance. Only for the Supernova DW LOCo outperforms LOCo in the early stage of the optimization, which is surprising to me, since the collisions of well-performing configuration should happen in later stages of the optimization process. This is somewhat contradicting the intuition provided by the authors.

- How many independent trials do you conduct for each method in Figure 3?

- Overall, Figure 3 might be somewhat missleading. If I am not mistaken you have to first sample several hundred data points to train the neural networks, which is not necessary for  TPE, PSO and BO. This initial phase is not reflecting in Figure 3 and puts these method into a disadvantage.

- Looking at Figure 9 it seems that LOCo is quite sensitive to the choice of lambda. For non-weighted version of LOCo lambda = 0.1 seems to work best, however for DW LOCo this choice of lambda performs worst and larger lambda (i.e 10) seem to work better.

- I am missing additional ablation studies for the other hyperparameters: \zeta and \rho



## Minor Comments

- Typo in Section 4.2: yi is defined twice instead of yj

- Increase font size of the legend in Figure 3

- Increase the line width in Figure 3



**Summary Of The Paper:**

Despite its success, Gaussian process based Bayesian optimization still struggles in high dimensional search spaces. Current approaches aim to learn an embedding to optimize the objective in a low dimensional continuous latent space. This paper provides evidence that with current approaches, different data points in the input space can be mapped to same point in the latent space. To avoid these collisions, the paper proposes a new regularization technique based on pairs of observed datapoints.

**Summary Of The Review:**

Overall I think the paper tackles an important problem and provides a sensible solution to it. However, I am underwhelmed by the empirical results of the paper and not yet convinced that the proposed approach actually works reliably in practice.

---

### Decision · Program_Chairs · 2022-01-20

**Decision:**

Reject

**Comment:**

In this paper, the problem of identifying a low-dimensional latent space for high-dimensional Bayesian optimization (BO) is considered. In particular, the authors focus on the problem of collision, where different points in the original space become identical in the latent space, and propose a regularization method to avoid this problem. Latent space identification for high-dimensional Bayesian optimization is an interesting and the authors' approach sounds reasonable. However, many reviewers pointed out that the discussion and results in the paper do not provide sufficient evidence for the authors' claims. Therefore, we have to conclude that the paper cannot be accepted at this time.